# Joint Modeling of Spatial and Temporal Multiscales in Molecular Dynamics

## Abstract

Molecular dynamics simulations are crucial for understanding complex physical, chemical, and biological processes at the atomic level. However, accurately capturing interactions across multiple spatial and temporal scales remains a significant challenge. We present a novel framework that jointly models spatial and temporal multiscale interactions in molecular dynamics. Our approach leverages Graph Fourier Transforms to decompose molecular structures into different spatial scales and employs Neural Ordinary Differential Equations to model the temporal dynamics at each scale. This unified framework explicitly links spatial structures with temporal evolution, enabling more accurate and comprehensive simulations of molecular systems. We evaluate our model on the MD17 and alanine dipeptide datasets, demonstrating consistent performance improvements over state-of-the-art baselines across multiple molecules. Importantly, we introduce physics-based evaluation metrics, including bond lengths, bond angles, and dihedral angles, to assess the practical effectiveness of our model in preserving molecular geometry over long time horizons. Ablation studies confirm the significant contributions of both spatial and temporal multiscale modeling components. Our method advances the simulation of complex molecular systems, potentially accelerating research in computational chemistry, drug discovery, and materials science.

## 1 Introduction

Molecular dynamics (MD) simulations are indispensable tools for investigating the behavior of molecular systems at the atomic level, offering profound insights into physical (Bear & Blaisten-Barojas, 1998), chemical (Wang et al., 2011), and biological (Salo-Ahen et al., 2020) processes. These simulations must capture interactions occurring across a wide range of spatial and temporal scales—from localized bond vibrations to long-range non-bonded interactions—posing significant computational challenges. Accurately modeling these multi-scale interactions requires sophisticated methods that can efficiently represent complex dependencies among atoms (Vakis et al., 2018).

An intrinsic link exists between the spatial scales of molecular interactions and their corresponding temporal dynamics. To illustrate this connection, consider the vibrational spectrum of a water molecule, a well-studied system in molecular spectroscopy. In Fourier-transform infrared (FTIR) spectroscopy (Griffiths, 1983), different absorption peaks correspond to distinct vibrational modes, each associated with specific spatial and temporal scales. As shown in Figure 1(a), high-frequency absorption peaks arise from localized O-H stretching vibrations, involving interactions over short spatial ranges and resulting in fast temporal dynamics. Conversely, low-frequency peaks correspond to collective motions like hydrogen bonding and bending modes, which involve longer spatial ranges and exhibit slower temporal dynamics. This spectrum exemplifies how localized spatial interactions correspond to faster temporal dynamics, while extended spatial interactions lead to slower dynamics.

Existing methods have attempted to address multi-scale interactions in MD simulations but often focus on either spatial or temporal scales independently. Coarse-grained techniques (Joshi & Deshmukh, 2021; Wang et al., 2019) reduce degrees of freedom to capture larger spatial scales but may lose important high-frequency details inherent in localized interactions. Neural operator-based methods like ITO (Schreiner et al., 2024), TimeWarp (Klein et al., 2024), and EGNO (Xu et al., 2024) are effective in extracting different temporal scales of signals but do not explicitly model the varying spatial scales that give rise to these temporal dynamics. Similarly, graph neural ODEs such

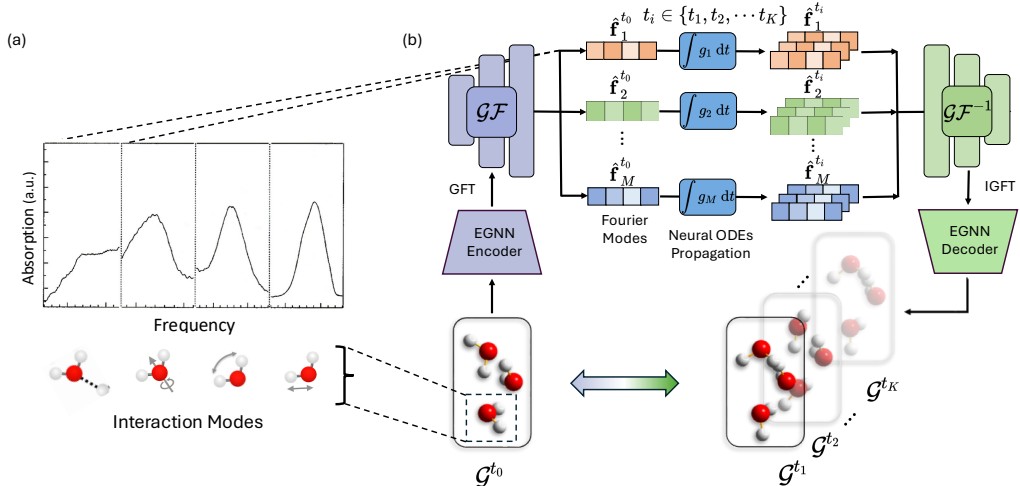

Figure 1: Overview of our framework for joint modeling of spatial and temporal multiscales in molecular dynamics. (a) FTIR spectrum of a water molecule, showing absorption peaks at different frequencies corresponding to various vibrational modes (Brubach et al., 2005). High-frequency peaks are associated with localized vibrations (e.g., O-H stretching), while low-frequency peaks correspond to global motions (e.g., bending modes). (b) Our model decomposes the molecular graph into different spatial scales using Graph Fourier Transform and models the temporal dynamics at each scale using Neural ODEs, capturing the correlation between spatial and temporal multiscale interactions.

as NDCN (Zang & Wang, 2020) and GG-ODE (Huang et al., 2023a) model temporal evolution in continuous time but often overlook the intrinsic connection between spatial structures and temporal dynamics.

The recognition of the aforementioned challenge naturally raises a pivotal question:

> *How to simultaneously model spatial and temporal multi-scale interactions in molecular dynamics for comprehensive and accurate simulations?*

In this work, we propose a novel approach that explicitly models the joint spatial and temporal multiscale interactions in molecular systems. Inspired by molecular spectroscopy techniques like FTIR, which reveal the dual connotations of frequencies in both spatial and temporal domains, we leverage Graph Fourier Transformation to decompose molecular interactions into different spatial scales. This transformation captures the high-frequency (localized) and low-frequency (extended) spatial interactions, effectively encoding the spatial multiscale nature of the molecule.

To model the temporal dynamics associated with each spatial scale, we employ Neural Ordinary Differential Equations (Neural ODEs) (Chen et al., 2018). The adaptive stepping nature of Neural ODEs allows for flexible and efficient modeling of dynamics at different time scales, accommodating both fast and slow temporal evolutions. By propagating the decomposed spatial scales through time using Neural ODEs, we capture the temporal dynamics corresponding to each spatial scale. After temporal propagation, we reconstruct the molecular state in the physical space by an inverse Graph Fourier Transformation.

This joint modeling approach is motivated by the natural association between spatial and temporal scales in molecular systems: localized spatial interactions (high-frequency spatial scales) correspond to faster temporal dynamics, while extended spatial interactions (low-frequency spatial scales) correspond to slower temporal dynamics. By explicitly linking these scales, our method provides a comprehensive framework for capturing the complex interplay between spatial structures and temporal dynamics in MD simulations.

By integrating spatial and temporal multiscale modeling, our approach advances the simulation of complex molecular systems. Extensive experiments on molecular dynamics datasets demonstrate that our model outperforms baseline models, highlighting the effectiveness of capturing both spatial and temporal multiscale interactions.

Our contributions can be summarized as follows:

1. **A New Perspective on Multiscale Modeling**: We provide a novel viewpoint by explicitly linking spatial and temporal scales in molecular dynamics, addressing the multiscale challenge through joint modeling of spatial structures and temporal dynamics.

2. **A Novel Framework for Joint Multiscale Modeling**: We introduce a framework that utilizes Graph Fourier Transformation to decompose molecular interactions into different spatial scales and employs Neural ODEs to model the temporal dynamics at each scale. The adaptive stepping of Neural ODEs allows for flexible modeling of dynamics at various time scales.

3. **An Effective Model Architecture**: We design a new model architecture that implements our proposed framework, effectively capturing and propagating different spatial scales through time to model the multiscale nature of molecular systems. Extensive experiments and ablations show the effectiveness of our architecture.

## 2 RELATED WORK

We review relevant works on multi-scale modeling in molecular dynamics, focusing on coarse-grained methods, neural operator models, and graph neural ODEs.

### 2.1 COARSE-GRAINED MOLECULAR DYNAMICS

Coarse-grained molecular dynamics (CGMD) (Noid, 2013; Saunders & Voth, 2013) reduce computational complexity by grouping atoms into larger units or "beads," simplifying molecular representations. This approach effectively captures large-scale molecular motions over extended periods, facilitating the study of complex systems like protein folding and polymer dynamics. F3low (Li et al., 2024) extends CG modeling to the SE(3) manifold using flow-matching for autoregressive sampling, enabling broader exploration of conformational space and improved insights into protein folding. Two for One (Arts et al., 2023) uses score-based generative models to learn CG force fields without force inputs, simplifying training and improving performance in protein simulations while preserving dynamics. However, CGMD primarily addresses spatial scaling and often sacrifices fine-grained details inherent in localized interactions, such as bond vibrations and stretching. Moreover, it lacks explicit mechanisms for modeling temporal multi-scale dynamics, limiting its ability to capture the interplay between fast and slow processes in molecular systems.

### 2.2 NEURAL OPERATOR MODELS

Neural operator models (Kovachki et al., 2023) have emerged as powerful tools for learning mappings between infinite-dimensional function spaces, showing promise in modeling complex dynamic processes. Conditional Diffusion Model (Hsu et al., 2024) introduces score dynamics (SD), which learns accelerated evolution operators from molecular simulations using diffusion models, enabling larger timesteps and faster simulations while maintaining accuracy. In molecular dynamics, methods like Implicit Transfer Operator (ITO) (Schreiner et al., 2024), Timewarp (Klein et al., 2024), and Equivariant Graph Neural Operator (EGNO) (Xu et al., 2024) aim to capture temporal dynamics over multiple scales. ITO and Timewarp employ coarse-graining and time-stepping techniques to accelerate simulations but focus primarily on temporal scaling, potentially losing critical spatial details. EGNO models temporal evolution using neural operators with SE(3) equivariance but does not explicitly handle spatial multi-scale interactions. These methods often treat spatial and temporal scales separately, lacking a joint modeling framework that captures their intrinsic connection.

## 2.3 GRAPH NEURAL ODE MODELS

Graph Neural ODEs integrate Graph Neural Networks into continuous-time dynamics using Neural ODE frameworks (Chen et al., 2018), enabling the modeling of time-evolving graph-structured data (Zang & Wang, 2020; Huang et al., 2023a;b). While effective in capturing temporal evolution on graphs, these models typically rely on local message passing, which may not adequately represent multi-scale spatial interactions present in molecular systems. Moreover, they often do not explicitly link spatial scales with temporal dynamics, potentially overlooking the relationship between localized spatial interactions and fast temporal dynamics, as well as extended spatial interactions and slow dynamics.

## 3 PROBLEM DEFINITION

For the task of molecular dynamics, we aim to predict the future conformations of a molecular system based on its initial structure. Let $\mathcal{G}^{t_0} = (\mathcal{V}, \mathcal{E})$ represent the molecular graph at the initial time $t_0$, where $\mathcal{V} = \{v_1, v_2, \ldots, v_N\}$ is the set of $N$ atoms in the molecule, and $\mathcal{E} \subseteq \mathcal{V} \times \mathcal{V}$ is the set of optional edges representing chemical bonds or interactions between atoms.

The position of each atom $v_i \in \mathcal{V}$ at time $t_0$ is given by its 3D coordinates $\mathbf{x}_i^{t_0} \in \mathbb{R}^3$. Our objective is to predict the atomic positions $\mathbf{X}(t_k) := \{\mathbf{x}_1^{t_k}, \ldots, \mathbf{x}_N^{t_k}\}$ at future time points $t_1, t_2, \ldots, t_K$, where $t_1, t_2, \ldots, t_K$ are irregularly sampled on a continuous time scale. All data are parallel sequences from molecular systems of the same type, meaning the data contain conformations sampled from similar molecular systems.

## 4 METHODOLOGY

In this section, we introduce our framework for jointly modeling spatial and temporal multiscale interactions in molecular dynamics. Our approach integrates Equivariant Graph Neural Networks (EGNNs) (Satorras et al., 2021) with Graph Fourier Transforms (GFTs) and Neural Ordinary Differential Equations (Neural ODEs) (Chen et al., 2018).

### 4.1 EQUIVARIANT GRAPH NEURAL NETWORK ENCODER

We start by constructing a molecular graph $\mathcal{G} = (\mathcal{V}, \mathcal{E})$, where each node $i \in \mathcal{V}$ represents an atom with features $\mathbf{h}_i$ and position $\mathbf{x}_i \in \mathbb{R}^3$. The edges $\mathcal{E}$ represent bonds or interactions between atoms.

Our EGNN encoder processes the molecular graph to produce updated node features and positions that are equivariant to rotations and translations.

Specifically, the EGNN updates node features and positions through message passing:

$$\mathbf{m}_{ij} = \phi_e(\mathbf{h}_i, \mathbf{h}_j, \mathbf{x}_i - \mathbf{x}_j), \tag{1}$$

$$\mathbf{h}_i' = \phi_h(\mathbf{h}_i, \sum_{j \in \mathcal{N}(i)} \mathbf{m}_{ij}), \tag{2}$$

$$\mathbf{x}_i' = \mathbf{x}_i + \frac{1}{|\mathcal{N}(i)|} \sum_{j \in \mathcal{N}(i)} \psi(\mathbf{m}_{ij})(\mathbf{x}_i - \mathbf{x}_j), \tag{3}$$

where $\phi_e$, $\phi_h$, and $\psi$ are learnable functions, and $\mathcal{N}(i)$ denotes the neighbors of node $i$.

In the above equations, $\mathbf{h}_i$ are scalar latent features (rotationally invariant), and $\mathbf{x}_i$ are vectorial features (rotationally equivariant). The updates are carefully designed to ensure that the updated positions $\mathbf{x}_i'$ maintain translational and rotational equivariance, as the position updates depend only on relative positions $\mathbf{x}_i - \mathbf{x}_j$.

## 4.2 GRAPH FOURIER TRANSFORM

To capture multiscale spatial interactions, we apply the Graph Fourier Transform (GFT) to the node features and positions.

The GFT of the node features $\mathbf{H} \in \mathbb{R}^{N \times F}$ is given by:

$$\tilde{\mathbf{H}} = \mathbf{U}^\top \mathbf{H}, \tag{4}$$

where $\mathbf{U}$ is the matrix of eigenvectors of the graph Laplacian $\mathbf{L}$.

Similarly, the GFT of the positions $\mathbf{X} \in \mathbb{R}^{N \times 3}$ is:

$$\tilde{\mathbf{X}} = \mathbf{U}^\top \mathbf{X}. \tag{5}$$

To preserve translational invariance, we note that the first eigenvector of the graph Laplacian corresponds to the zero eigenvalue and represents the DC component (uniform translation). By centering the positions $\mathbf{X}$ before applying the GFT, i.e., $\mathbf{X}_c = \mathbf{X} - \frac{1}{N} \sum_{i=1}^{N} \mathbf{x}_i$, we ensure that the GFT captures only the variations in positions, which are invariant to translations.

## 4.3 NEURAL ODEs FOR TEMPORAL DYNAMICS

Each frequency component from the GFT corresponds to a spatial scale. We model the temporal evolution of each component using Neural ODEs:

$$\frac{d\tilde{\mathbf{H}}_k(t)}{dt} = f_\theta(\tilde{\mathbf{H}}_k(t), t), \quad \forall k, \tag{6}$$

where $\tilde{\mathbf{H}}_k$ denotes the $k$-th frequency component, and $f_\theta$ is a neural network parameterized by $\theta$.

Similarly, for positions:

$$\frac{d\tilde{\mathbf{X}}_k(t)}{dt} = g_\phi(\tilde{\mathbf{X}}_k(t), t), \quad \forall k. \tag{7}$$

Importantly, by integrating the Neural ODEs in the spectral domain and ensuring that the DC component is handled appropriately, we maintain translational equivariance in the reconstructed positions.

## 4.4 INVERSE GRAPH FOURIER TRANSFORM

After evolving the frequency components over time, we reconstruct the node features and positions using the Inverse Graph Fourier Transform (IGFT):

$$\mathbf{H}(t) = \mathbf{U}\tilde{\mathbf{H}}(t), \tag{8}$$
$$\mathbf{X}_c(t) = \mathbf{U}\tilde{\mathbf{X}}(t). \tag{9}$$

We then recover the absolute positions by adding back the mean position:

$$\mathbf{X}(t) = \mathbf{X}_c(t) + \mathbf{x}_{\text{mean}}, \tag{10}$$

where $\mathbf{x}_{\text{mean}}$ is the mean position computed earlier. This ensures that the reconstructed positions maintain the correct translational behavior.

By carefully handling the mean and the DC component, we ensure that the entire process from EGNN to GFT, Neural ODEs, and IGFT preserves translational equivariance.

## 4.5 Equivariant Graph Neural Network Decoder

Finally, we refine the reconstructed features and positions using an EGNN decoder, which has a similar structure to the encoder.

$$\mathbf{m}'_{ij} = \phi'_e(\mathbf{h}_i(t), \mathbf{h}_j(t), \mathbf{x}_i(t) - \mathbf{x}_j(t)), \tag{11}$$

$$\mathbf{h}''_i = \phi'_h(\mathbf{h}_i(t), \sum_{j \in \mathcal{N}(i)} \mathbf{m}'_{ij}), \tag{12}$$

$$\mathbf{x}''_i = \mathbf{x}_i(t) + \frac{1}{|\mathcal{N}(i)|} \sum_{j \in \mathcal{N}(i)} \psi'(\mathbf{m}'_{ij})(\mathbf{x}_i(t) - \mathbf{x}_j(t)). \tag{13}$$

Again, the updates depend only on relative positions, ensuring that translational equivariance is maintained throughout the decoding process.

## 4.6 Loss Function

To train the model, we use the Mean Squared Error (MSE) loss, which measures the difference between the predicted atomic positions and the ground truth positions at each predicted time point. Given that the goal is to predict the molecular conformations at time points $t_1, t_2, \ldots, t_K$, the MSE is calculated as follows:

Let $\mathbf{x}_i^{t_j} \in \mathbb{R}^3$ be the ground truth 3D coordinates of atom $i$ at time $t_j$, and $\tilde{\mathbf{x}}_i^{t_j} \in \mathbb{R}^3$ be the predicted coordinates for the same atom at time $t_j$. The MSE loss is defined as:

$$\mathcal{L}_{\text{MSE}} = \frac{1}{NK} \sum_{j=1}^{K} \sum_{i=1}^{N} \left\| \mathbf{x}_i^{t_j} - \tilde{\mathbf{x}}_i^{t_j} \right\|_2^2, \tag{14}$$

where $N$ is the number of atoms and $K$ is the number of time points.

This loss encourages the model to minimize the Euclidean distance between the predicted and actual atomic positions across all time steps, ensuring accurate trajectory prediction for the molecular system.

The MSE loss is applied at each time point, thus aligning the predicted future states with the true molecular dynamics trajectory. The model is trained by minimizing $\mathcal{L}_{\text{MSE}}$ over all predicted time points.

## 5 Experiments

In this section, we present a comprehensive evaluation of our proposed model on molecular dynamics datasets, comparing its performance against several baseline methods and conducting ablation studies to assess the contributions of different model components.

Our experiments are designed to address the following research questions:

- **RQ1**: Does jointly modeling spatial and temporal multiscale interactions improve prediction accuracy?
- **RQ2**: How do the different components of our model (the spatial multiscale modeling and the temporal multiscale modeling) contribute to its performance?
- **RQ3**: Does extracting different scales stabilize predictions over longer periods of time?
- **RQ4**: Does our model capture the latent correlation between specific spatial scales and specific temporal scales?

## 5.1 Dataset

We evaluate our model using the MD17 dataset (Chmiela et al., 2017), which contains molecular dynamics trajectories for eight small molecules: Aspirin, Benzene, Ethanol, Malonaldehyde, Naphthalene, Salicylic Acid, Toluene, and Uracil. The dataset provides atomic trajectories simulated

under quantum mechanical forces, capturing realistic molecular motions. We also evaluated our model on the alanine dipeptide dataset (Schreiner et al., 2024), a standard benchmark for studying conformational dynamics in proteins. Our task is to predict the future positions of atoms given the initial state of the molecular system.

## 5.2 EXPERIMENTAL SETUP

For each molecule, we partition the trajectory data into training, validation, and test sets, using 500 samples for training, 2000 for validation, and 2000 for testing. The time scope $\Delta T$ for each piece of data is set to 3000 simulation steps, providing a challenging prediction horizon.

A key aspect of our experimental setup is the use of **irregular timestep sampling**, in contrast to the equi-timestep sampling used in some baseline models like EGNO, to better mimic the variable time intervals in real-world physical systems. This setting tests the models' ability to handle irregular temporal data. Nevertheless, we also provide evaluations based on equi-timestep sampling in the appendix for completeness.

Further details on the experimental setup, including hyperparameter choices and implementation specifics, are also provided in the uploaded codebase to OpenReview. Detailed specification can also be found in the appendix.

## 5.3 BASELINE MODELS

We compare our model against several state-of-the-art approaches:

- **NDCN** (Zang & Wang, 2020): A Graph Neural ODE model that integrates graph neural networks into the ODE framework to learn continuous-time dynamics of networked systems.
- **EGNN** (Satorras et al., 2021): An Equivariant Graph Neural Network that models molecular systems using 3D equivariant message passing but without explicit time propagation.
- **EGNO** (Xu et al., 2024): An Equivariant Graph Neural Operator that captures temporal dynamics using neural operators with regular timesteps.
- **ITO** (Schreiner et al., 2024): An Implicit Time-stepping Operator that integrates differential equations into the learning process for temporal evolution.

These baselines represent a range of approaches for modeling molecular dynamics, including methods that focus on either spatial modeling (EGNN), or temporal modeling (NDCN, EGNO, ITO).

## 5.4 RESULTS AND ANALYSIS

To address **RQ1**, we evaluate the performance of our model and the baselines on the MD17 dataset. The test Mean Squared Error (MSE) on the eight molecules for our model and the baseline methods are summarized in Table 1. For completeness, we also provide results in equi-timestep setting in Table 4 in the Appendix.

From Table 1, we observe that our model consistently outperforms the baseline methods across all eight molecules under irregular timestep sampling. This demonstrates the effectiveness of our approach in jointly modeling spatial and temporal multiscale interactions. The performance gains are particularly significant for molecules like Benzene and Aspirin, where our model reduces the MSE by substantial margins compared to the baselines.

The inferior performance of EGNO and ITO under irregular timestep sampling highlights their limitations in handling variable time intervals due to their reliance on discrete Fourier transformations in the temporal domain, which assume equi-timesteps. In contrast, EGNN, which, in our implementation, incorporates timestamp embeddings into a 3D GNN model, performs better than EGNO and ITO in this setting but still falls short of our model. This underscores the advantage of our method in modeling dynamics with irregular timesteps by explicitly capturing multiscale interactions.

Table 1: MSE ($\times 10^{-2}$ Å$^2$) on MD17 dataset, timesteps sampled with **irregular** intervals. Upper part: comparison between the baselines and our model. Best results are marked in **bold**, and second-to-best results are underlined. Lower part: results for ablation models. Compared with the base model performance, deteriorated results are marked in orange, while improved results are marked in green.

| | Aspirin | Benzene | Ethanol | Malonaldehyde | Naphthalene | Salicylic | Toluene | Uracil |
|---|---|---|---|---|---|---|---|---|
| NDCN | $29.75_{\pm0.02}$ | $70.13_{\pm0.98}$ | $10.05_{\pm0.02}$ | $42.28_{\pm0.07}$ | $2.30_{\pm0.00}$ | $3.43_{\pm0.05}$ | $12.33_{\pm0.00}$ | $2.39_{\pm0.00}$ |
| EGNN | $9.09_{\pm0.10}$ | $\underline{49.15}_{\pm1.68}$ | $4.46_{\pm0.01}$ | $\underline{12.52}_{\pm0.05}$ | $0.40_{\pm0.02}$ | $0.89_{\pm0.01}$ | $8.98_{\pm0.09}$ | $0.64_{\pm0.00}$ |
| EGNO | $10.60_{\pm0.01}$ | $52.53_{\pm2.40}$ | $4.52_{\pm0.06}$ | $12.89_{\pm0.06}$ | $0.46_{\pm0.01}$ | $1.07_{\pm0.00}$ | $9.31_{\pm0.10}$ | $0.67_{\pm0.01}$ |
| ITO | $12.74_{\pm0.10}$ | $57.84_{\pm0.86}$ | $7.23_{\pm0.00}$ | $19.53_{\pm0.01}$ | $1.77_{\pm001}$ | $2.53_{\pm0.03}$ | $9.96_{\pm0.04}$ | $1.71_{\pm0.15}$ |
| Ours | $\mathbf{8.85}_{\pm0.02}$ | $\mathbf{40.86}_{\pm0.98}$ | $\mathbf{4.41}_{\pm0.06}$ | $\mathbf{12.49}_{\pm0.00}$ | $\mathbf{0.40}_{\pm0.01}$ | $\mathbf{0.87}_{\pm0.01}$ | $\mathbf{8.63}_{\pm0.04}$ | $\mathbf{0.62}_{\pm0.02}$ |
| w/o Fourier | $8.79_{\pm0.06}$ | $46.23_{\pm3.16}$ | $4.42_{\pm0.01}$ | $12.50_{\pm0.03}$ | $0.46_{\pm0.00}$ | $0.95_{\pm0.07}$ | $8.70_{\pm0.01}$ | $0.61_{\pm0.01}$ |
| w/o ODE | $8.93_{\pm0.01}$ | $43.09_{\pm2.07}$ | $4.42_{\pm0.00}$ | $12.49_{\pm0.01}$ | $0.41_{\pm0.01}$ | $0.88_{\pm0.02}$ | $8.65_{\pm0.08}$ | $0.62_{\pm0.02}$ |

To further validate the effectiveness of our approach on more complex molecular dynamics scenarios, we extend our experiments to include the **alanine dipeptide** dataset, a standard benchmark in molecular dynamics simulations Schreiner et al. (2024). This dataset captures essential aspects of protein backbone dynamics and presents a greater challenge due to its conformational flexibility and rich torsional dynamics.

**Experimental Results on Alanine Dipeptide**  *Units and Calculations:* Note that the alanine dipeptide dataset operates in nanometers (nm), while the MD17 dataset uses angstroms (Å). When calculating bond lengths and bond angles, we consider all heavy atoms (excluding hydrogen atoms) to focus on the primary structural framework of the molecule.

The Mean Squared Error (MSE) results on the alanine dipeptide dataset are summarized in Table 2.

Table 2: Mean Squared Error (MSE) ($\times 10^{-3}$ nm$^2$) on the alanine dipeptide dataset. Best results are marked in **bold**.

| Model | MSE ($\times 10^{-3}$ nm$^2$) |
|---|---|
| NDCN | $12.27 \pm 0.19$ |
| ITO | $26.95 \pm 0.19$ |
| EGNO | $6.92 \pm 0.26$ |
| EGNN | $5.67 \pm 0.08$ |
| **Ours** | $\mathbf{5.24 \pm 0.19}$ |

From Table 2, we observe that our model achieves the lowest MSE among all compared methods, with approximately 7.5% reduction in error compared to EGNN. This indicates a significant improvement over the baselines and demonstrates the effectiveness of our approach on more complex molecular dynamics data.

**In-depth Analysis of Molecular Structure Recovery**  To provide a more comprehensive evaluation, we analyzed the bond lengths and bond angles predicted by our model compared to the ground truth. Table 3 presents the Mean Absolute Errors (MAEs) and relative errors for bond lengths and bond angles.

Table 3: Mean Absolute Errors (MAEs) and relative errors for bond lengths and bond angles on the alanine dipeptide dataset. Best results are marked in **bold**.

| Model | Bond Length MAE (nm) | Rela. Err. (%) | Bond Angle MAE (°) | Rela. Err. (%) |
|---|---|---|---|---|
| EGNN | $0.0209 \pm 0.0006$ | $15.32 \pm 0.49$ | $12.44 \pm 0.91$ | $10.48 \pm 0.76$ |
| EGNO | $0.0229 \pm 0.0018$ | $16.75 \pm 1.23$ | $10.54 \pm 0.11$ | $8.89 \pm 0.11$ |
| **Ours** | $\mathbf{0.0188 \pm 0.0022}$ | $\mathbf{13.74 \pm 1.66}$ | $\mathbf{10.47 \pm 1.03}$ | $\mathbf{8.80 \pm 0.89}$ |

Our model achieves the lowest MAEs and relative errors in both bond lengths and bond angles, demonstrating its superior ability to accurately recover internal molecular structures compared to the baselines.

**Ramachandran Plot Analysis**   We further evaluate the ability of our model to capture torsional dynamics by analyzing the distribution of dihedral (torsion) angles $\phi$ (phi) and $\psi$ (psi). Figure 2 presents the Ramachandran plots comparing the predicted and ground truth distributions.

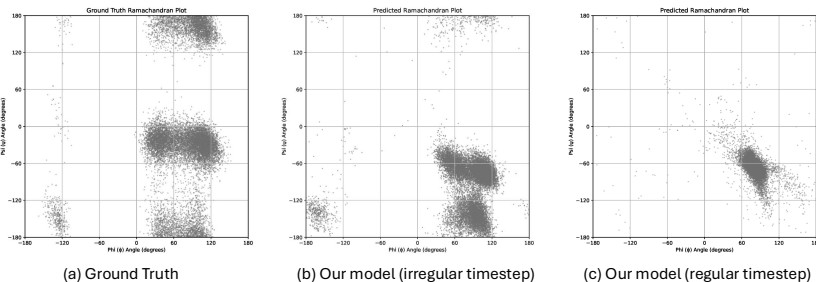

(a) Ground Truth          (b) Our model (irregular timestep)          (c) Our model (regular timestep)

Figure 2: Ramachandran plots showing the distribution of dihedral angles $\phi$ and $\psi$ for alanine dipeptide. (a) Ground truth distribution. (b) Distribution predicted by our model, trained on irregular timestep sampling setting; (c) trained on regular timestep sampling setting.

As shown in Figure 2, our model accurately captures the characteristic conformational regions of alanine dipeptide, closely matching the ground truth distributions. This indicates that our model effectively models the torsional dynamics, which are critical for understanding protein folding and conformational changes. We also observed that the model trained with irregular timestep sampling better recovers the characteristic patterns in the Ramachandran plot, indicating an improved ability to learn the molecule's conformational space by incorporating the conformations at multiple time scales.

These additional experiments on alanine dipeptide further confirm the robustness and effectiveness of our approach in modeling complex molecular dynamics with irregular timesteps, outperforming existing state-of-the-art methods.

### 5.5   ABLATION STUDIES

To address **RQ2**, we conduct ablation studies to assess the contributions of the key components in our model: the Graph Fourier Transformation for spatial multiscale modeling and the Neural ODE for temporal multiscale modeling. We consider two ablated versions of our model:

- **w/o Fourier**: The model without the Graph Fourier Transformation. In this variant, we directly use the spatial features without decomposing them into different spatial scales.
- **w/o ODE**: The model without the Neural ODE component. In this variant, we replace the Neural ODE with a standard feed-forward neural network for temporal propagation.

The results of the ablation studies are presented in the lower part of Table 1. The results for the equi-timestep setting is provided in the lower part of Table 4 in the Appendix.

From the ablation results, we observe that removing either the Graph Fourier Transformation or the Neural ODE component leads to a degradation in performance on most molecules. For instance, without the Fourier component, the MSE on Benzene increases from 40.86 to 46.23, and without the ODE component, it increases to 43.09. This indicates that both the spatial multiscale modeling via Graph Fourier Transformation and the temporal multiscale modeling via Neural ODE contribute significantly to the overall performance of the model.

In some cases, the ablated models perform comparably to the full model (e.g., on Aspirin and Uracil). However, the full model consistently achieves the best overall performance across all molecules, confirming the importance of jointly modeling both spatial and temporal multiscales.

## 5.6 Long-Term Prediction Stability

To address **RQ3**, we evaluate the performance of our model and the baselines on longer-term predictions. We test the models on predicting atomic positions over extended time horizons of 1000, 2000, 3000, 4000, and 5000 simulation steps on two representative molecules: Benzene and Malonaldehyde. The results are shown in Figure 3.

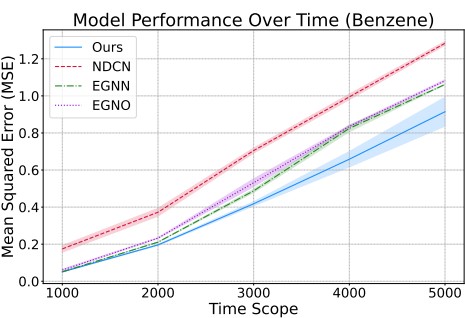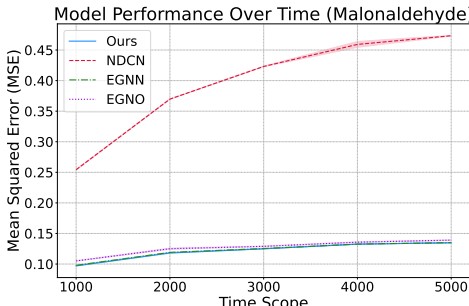

Figure 3: Long-term prediction MSE over different time horizons for Benzene and Malonaldehyde.

From Figure 3, we observe that our model maintains superior performance and demonstrates more stable predictions over longer periods compared to the baselines. The error growth rate of our model is less steep, indicating that it is better at capturing the dynamics that govern the system over extended time scales. This can be attributed to our model's ability to identify and propagate the lower-frequency modes, which carry more global information and are crucial for long-term predictions. The baselines, lacking explicit multiscale modeling, tend to accumulate errors more rapidly over time.

## 5.7 Case Study: Correlation between Spatial and Temporal Scales

To address **RQ4**, we conduct a case study on the Benzene molecule to investigate whether our model captures the latent correlation between specific spatial scales and temporal scales. We analyze the Fourier coefficients of different spatial modes over time to observe how they evolve.

As shown in Figure 4, the lower spatial modes (e.g., mode 0), which represent global, low-frequency spatial scales on the graph, also encode temporal signals of lower frequency. These modes change gradually over time, reflecting slower dynamics associated with extended spatial interactions. Conversely, higher spatial modes (e.g., mode 3), corresponding to more localized spatial interactions, has a significantly lower portion for the lower frequency temporal signals, exhibit higher-frequency temporal dynamics with rapid fluctuations. This demonstrates that our model effectively captures the inherent correlation between specific spatial scales and their associated temporal scales, validating our approach of jointly modeling spatial and temporal multiscale interactions.

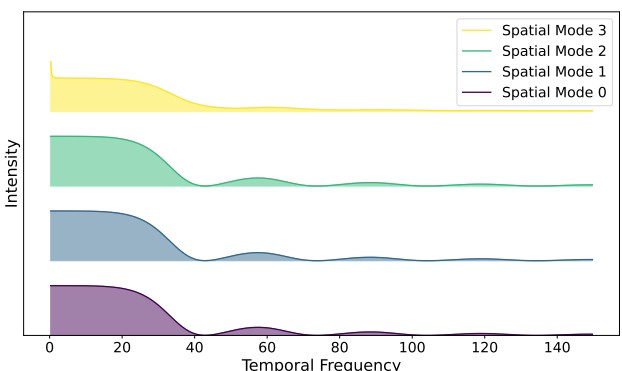

Figure 4: Fourier coefficients of different spatial modes over time for Benzene. Lower spatial modes (e.g., mode 0) correspond to a higher portion of lower-frequency temporal signals, while higher spatial modes (e.g., mode 3) exhibit higher-frequency temporal dynamics. (x axis unit: $ps^{-1}$)

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

## A  ADDITIONAL EXPERIMENT RESULTS

Table 4: MSE ($\times 10^{-2}$) on MD17 dataset, timesteps sampled with **equal** intervals. Upper part: comparison between the baselines and our model. Best results are marked in **bold**, and second-to-best results are underlined. Lower part: results for ablation models. Compared with the base model performance, deteriorated results are marked in orange, while improved results are marked in green.

| | Aspirin | Benzene | Ethanol | Malonaldehyde | Naphthalene | Salicylic | Toluene | Uracil |
|---|---|---|---|---|---|---|---|---|
| NDCN | $31.73_{\pm0.40}$ | $56.21_{\pm0.30}$ | $10.74_{\pm0.02}$ | $46.55_{\pm0.28}$ | $2.25_{\pm0.01}$ | $3.58_{\pm0.11}$ | $13.92_{\pm0.02}$ | $2.38_{\pm0.00}$ |
| EGNN | $\underline{9.24}_{\pm0.07}$ | $57.85_{\pm2.70}$ | $\underline{4.63}_{\pm0.00}$ | $\mathbf{12.81}_{\pm0.01}$ | $\underline{0.38}_{\pm0.01}$ | $\mathbf{0.85}_{\pm0.00}$ | $\underline{10.41}_{\pm0.04}$ | $\underline{0.56}_{\pm0.02}$ |
| EGNO | $9.41_{\pm0.09}$ | $\underline{55.13}_{\pm3.21}$ | $\underline{4.63}_{\pm0.00}$ | $\mathbf{12.81}_{\pm0.01}$ | $0.40_{\pm0.01}$ | $0.93_{\pm0.01}$ | $10.43_{\pm0.10}$ | $0.59_{\pm0.01}$ |
| ITO | $20.56_{\pm0.03}$ | $57.25_{\pm0.58}$ | $8.60_{\pm0.27}$ | $28.44_{\pm0.73}$ | $1.82_{\pm0.17}$ | $2.48_{\pm0.34}$ | $12.47_{\pm0.30}$ | $1.33_{\pm0.12}$ |
| Ours | $\mathbf{9.16}_{\pm0.10}$ | $\mathbf{50.93}_{\pm6.28}$ | $\mathbf{4.62}_{\pm0.00}$ | $\underline{12.83}_{\pm0.00}$ | $\mathbf{0.35}_{\pm0.01}$ | $\underline{0.88}_{\pm0.05}$ | $\mathbf{10.31}_{\pm0.11}$ | $\mathbf{0.54}_{\pm0.01}$ |
| w/o Fourier | $8.79_{\pm0.26}$ | $51.28_{\pm5.38}$ | $4.62_{\pm0.00}$ | $12.81_{\pm0.01}$ | $0.37_{\pm0.01}$ | $0.90_{\pm0.03}$ | $10.35_{\pm0.10}$ | $0.60_{\pm0.01}$ |
| w/o ODE | $9.19_{\pm0.15}$ | $55.13_{\pm2.73}$ | $4.63_{\pm0.01}$ | $12.80_{\pm0.00}$ | $0.37_{\pm0.02}$ | $0.86_{\pm0.05}$ | $10.32_{\pm0.04}$ | $0.54_{\pm0.02}$ |

## B  EXPERIMENT SETUP

In this section, we provide a detailed description of the experimental setup for our model, including training procedures, model architecture, dataset inputs and outputs, hyperparameters, and an in-depth examination of how the input features $\mathbf{x}$ and $\mathbf{h}$ are transformed throughout the model. This information is based on the provided configuration file `config_md17_no.json` and the main training script `main_md17_no.py`.

### B.1  DATASETS

#### B.1.1  MD17 DATASET

- **Description**: The MD17 dataset consists of molecular dynamics simulations for various small organic molecules. It provides atomic trajectories computed using *ab initio* methods, capturing realistic molecular motions.
- **Molecule Used**: In this experiment, we focus on the **Alanine Dipeptide (ala2)** molecule, specified by `"mol": "ala2"` in the configuration.

- **Data Directory**: The data is expected to be located in the directory specified by `"data_dir": "ala"`.
- **Data Split**:
  - **Training Set**: 500 samples (as specified by `"max_training_samples": 500`).
  - **Validation Set**: 2,000 samples.
  - **Test Set**: 2,000 samples.
- **Time Steps**: Each sample contains **8 time steps** (`"num_timesteps": 8`), capturing the progression of the molecular state over time. The time interval between frames is determined by `"delta_frame": 3000`.
- **Uneven Sampling**: Disabled (`"uneven_sampling": false`), indicating that timesteps are sampled at regular intervals.

## B.2 MODEL ARCHITECTURE

Our model aims to jointly model spatial and temporal multiscale interactions in molecular dynamics simulations. The primary components of the model include:

### B.2.1 1. ENCODER (EQUIVARIANT GRAPH NEURAL NETWORK)

- **Purpose**: Encode the initial molecular structure and properties into latent representations.
- **Input Features**:
  - **Atomic Positions** ($\mathbf{x}$): 3D coordinates of atoms at each time step.
  - **Atomic Features** ($\mathbf{h}$): Scalars representing properties like atom types or initial embeddings.
- **Process**: Utilize an **Equivariant Graph Neural Network (EGNN)** to encode spatial relationships while preserving equivariance under rotations and translations. Update $\mathbf{h}$ and $\mathbf{x}$ based on neighboring information.

### B.2.2 2. GRAPH FOURIER TRANSFORM (GFT)

- **Purpose**: Decompose the molecular graph into different spatial frequency components to capture multiscale spatial interactions.
- **Process**: Apply GFT to the latent features $\mathbf{h}$ and positions $\mathbf{x}$ separately, transforming them into the spectral domain. The number of modes is specified by `"num_modes": 2`, indicating that we capture two spatial frequency components.

### B.2.3 3. NEURAL ORDINARY DIFFERENTIAL EQUATIONS (NEURAL ODES)

- **Purpose**: Model the temporal evolution of each frequency component obtained from the GFT.
- **Process**: For each frequency mode, we define a Neural ODE that models the continuous-time dynamics. The ODE solver used is `"solver": "dopri5"`, with relative and absolute tolerances specified by `"rtol": 1e-3` and `"atol": 1e-4`, respectively. Temporal embeddings are incorporated via `"time_emb_dim": 32`.

### B.2.4 4. INVERSE GRAPH FOURIER TRANSFORM (IGFT)

- **Purpose**: Transform the evolved frequency components back to the spatial domain to reconstruct the updated positions and features.

### B.2.5 5. DECODER (EQUIVARIANT GRAPH NEURAL NETWORK)

- **Purpose**: Refine the reconstructed features and produce the final predicted atomic positions.
- **Process**: Apply an EGNN similar to the encoder to update $\mathbf{h}$ and $\mathbf{x}$ based on the evolved features.

## B.3 HYPERPARAMETERS

The hyperparameters used in the experiment are as follows (from `config_md17_no.json`):

- **Experiment Name**: `"exp_name": "md17_exp"`
- **Number of Time Steps**: `"num_timesteps": 8`
- **Batch Size**: `"batch_size": 50`
- **Epochs**: `"epochs": 5000`
- **Use CUDA**: `"no_cuda": false`
- **Random Seed**: `"seed": 53`
- **Learning Rate**: `"lr": 0.0001`
- **Hidden Feature Dimension**: `"nf": 64`
- **Model Type**: `"model": "fourier"`
- **Attention Mechanism**: `"attention": 0`
- **Number of Layers**: `"n_layers": 5`
- **Maximum Training Samples**: `"max_training_samples": 500`
- **Data Directory**: `"data_dir": "ala"`
- **Normalize Differences**: `"norm_diff": false`
- **Learnable Fourier Basis**: `"learnable": false`
- **Weight Decay**: `"weight_decay": 1e-15`
- **Use Tanh Activation**: `"tanh": false`
- **Delta Frame**: `"delta_frame": 3000`
- **Molecule**: `"mol": "ala2"`
- **Time Embedding Dimension**: `"time_emb_dim": 32`
- **Number of Modes**: `"num_modes": 2`
- **ODE Solver**: `"solver": "dopri5"`
- **Relative Tolerance**: `"rtol": 1e-3`
- **Absolute Tolerance**: `"atol": 1e-4`
- **Uneven Sampling**: `"uneven_sampling": false`
- **Fourier Basis**: `"fourier_basis": "graph"`
- **Disable Fourier Block**: `"no_fourier": false`
- **Disable ODE Block**: `"no_ode": false`

## B.4 TRAINING PROCEDURE

- **Optimizer**: Adam optimizer with the specified learning rate and weight decay.
- **Learning Rate Scheduler**: StepLR scheduler with a step size and gamma (not specified in the config, but default values can be inferred from the code).
- **Loss Function**: Mean Squared Error (MSE) between predicted and true atomic positions.
- **Training Loop**:
  - Iterate over the specified number of epochs (`"epochs": 5000`).
  - For each batch, compute the loss and update model parameters via backpropagation.
  - Evaluate the model on the validation and test sets at regular intervals.

## C  BROADER IMPACT

In this paper, we introduced a novel framework for modeling molecular dynamics by jointly capturing spatial and temporal multiscale interactions. By leveraging the Graph Fourier Transform to decompose molecular structures into different spatial scales and employing Neural Ordinary Differential Equations to model the temporal dynamics at each scale, our approach explicitly links spatial structures with temporal evolution. Extensive experiments on the MD17 dataset demonstrated that our model consistently outperforms state-of-the-art baseline methods across multiple molecules, particularly under challenging conditions like irregular timestep sampling and long-term prediction horizons. Ablation studies confirmed the significant contributions of both the spatial multiscale modeling and the temporal multiscale modeling components. Our method provides a comprehensive and accurate representation of molecular behavior over time, advancing the simulation of complex molecular systems.

The proposed framework has the potential to significantly impact various domains by enhancing the accuracy and efficiency of molecular simulations. In computational chemistry and drug discovery, it can accelerate the understanding of molecular interactions and aid in the development of new therapeutics. In materials science, it can assist in designing novel materials with desired properties by accurately modeling molecular dynamics at different scales. Additionally, the method's computational efficiency may reduce resource requirements for large-scale simulations. However, it is important to consider potential limitations, such as the need for high-quality data and computational resources, as well as ethical considerations regarding the misuse of advanced simulation technologies. By addressing these factors, our work aims to contribute responsibly to the advancement of molecular science and technology.

