# OpenReview forum: "Joint Modeling of Spatial and Temporal Multiscales in Molecular Dynamics"
_ICLR.cc/2025/Conference — Submitted to ICLR 2025_

### Official Review · Reviewer_6VyJ · 2024-10-31

**Soundness:** 2
**Presentation:** 3
**Contribution:** 1
**Rating:** 3
**Confidence:** 5

**Summary:**

The paper presents a framework for generating molecular dynamics trajectory. The proposed model employs Graph Fourier Transformations (GFT), coupled with Neural ODEs to simulate the temporal dynamics at multiple scales. This joint modeling approach links localized, fast spatial interactions with high-frequency temporal dynamics and broader, slower interactions with low-frequency temporal changes. Evaluation is done on the MD17 dataset to reconstruct original MD trajectories in terms of MSE.

**Strengths:**

- The proposed method seems solid and has reasonable execution. The proposed model might be potentially suitable for modeling multi-scale spatiotemporal dynamics of a point cloud (with a reasonable choice of architecture and methodology).
- The presentation is clear and easy to follow.

**Weaknesses:**

- The method advancement is relatively incremental in the context of previous works, such as EGNO.
- The biggest issue is the evaluation is irrelevant to practical use cases of molecular modeling. Let me explain:

1. MD simulation is inherently chaotic. Fully recovering long-time trajectories in terms of MSE is not indicative of the practical usability of a model aiming at simulating molecular dynamics.
2. MD17 molecules are small, without interesting dynamics. Being able to simulate such small molecules is not indicative of the practical usability of the model on slightly more interesting molecules, such as dipeptides.

I understand the MD17 benchmark was used in previous works. However, I have to point out that in a paper titled "Joint Modeling of Spatial and Temporal Multiscale in Molecular Dynamics", the application matters. The MD17 MSE task on its own can only be seen as a toy, and flawed task, in the context of molecular dynamics modeling. In the context of modeling spatiotemporal dynamics, it might be OK to use it as a testbed, but not a very convincing benchmark. Some previous works applied their methods to the MD17 tasks, often with other spatio-temporal modeling tasks.

To this reviewer, the MD17 MSE task is of no practical relevance to the molecular simulation community, therefore the paper lacks evidence on the effectiveness of the proposed method on the task it aims to solve. A more reasonable task would be the experiments conducted in the ITO paper, which the authors used as a baseline. That is, simulating a slightly more complicated molecule such as a peptide or protein, and evaluating the model on its capability to recover ensemble properties.

Alternatively, the proposed method might shine on non-molecular spatiotemporal modeling tasks that are more relevant to practice.

**Questions:**

The paper lacks evidence that the proposed method solves a relevant molecular simulation task. The task proposed in the ITO paper, which the authors used as a baseline, would be a proper task. This reviewer would consider raising scores if evidence/insights on relevant task can be shown.

---

### Official Review · Reviewer_csJr · 2024-11-02

**Soundness:** 2
**Presentation:** 3
**Contribution:** 3
**Rating:** 5
**Confidence:** 4

**Summary:**

The paper introduces a framework for modeling molecular dynamics by jointly capturing spatial and temporal multiscale interactions. Leveraging Graph Fourier Transforms (GFT) for spatial decomposition and Neural Ordinary Differential Equations (Neural ODEs) for temporal propagation, the approach aims to enhance MD simulations. Evaluation on the MD17 dataset indicates improvements over baseline models, especially under irregular timestep sampling and extended prediction horizons. Overall, this paper proposes a promising concept for multiscale spatial and temporal prediction of MD trajectories. However, I recommend it as “marginally below the acceptance threshold” due to the issues outlined below.

**Strengths:**

* **Originality**: The integration of GFT and Neural ODEs to simultaneously model spatial and temporal scales in MD is a novel idea.
* **Significance**: If successfully executed, this approach could benefit fields that depend on molecular simulations by providing more comprehensive dynamics, potentially aiding in applications across computational chemistry and materials science.

**Weaknesses:**

* **Clarity**: The paper did not clearly articulate the methods, or there is still some confusion about the current methodologies.
* **Training Setup**: The training setups are missing in the paper.
* **Insufficiently Explained Experimental Design**: The paper does not clearly justify the experimental setup, for example, what does "irregular timestep sampling" exactly mean? How did the authors choose the irregular sampling?
* **Code not available**: I cannot review the codes of this work.
* **Physical soundness of the experiments**: Several aspects of the experimental setup appear to lack physical rigor. More details are listed  below

**Questions:**

Overall, I believe the authors present a promising approach for predicting MD trajectories. However, the following questions should be addressed to clarify key aspects and improve rigor:

1. **Soundness of the Method**: The authors claim in lines 45-46 that "*This spectrum exemplifies how localized spatial interactions correspond to faster temporal dynamics, while extended spatial interactions lead to slower dynamics.*" Yet, in Section 5.3 Neural Ordinary Differential Equations (Neural ODEs), the same timestep is applied across all ODEs, which seems inconsistent with this principle. Since localized and delocalized components correspond to different dynamical speeds, they should ideally use different timesteps.

    - This oversight also impacts the authors’ claim that the method supports multiscale temporal simulation. In reality, it supports irregular timestep simulation, but the timestep remains uniform across localized and delocalized Fourier components.

    - However, the claim of multiscale spatial simulation is valid.

2. **Clarity of Methodological Details**: Some parts of the methods section lack clarity and specificity.

    - The training setup for Eqs. 5 and 6 is not provided. Information on the types and dimensions of the neural networks used, as well as training protocols, would be helpful.

    - Eqs. 16 and 17 use the same notation as Eqs. 5 and 6. Do these equations share the same neural networks, or are they distinct? How were they trained?

3. **Typo**: There is a typo in lines 85-86.

4. **Performance Metrics and Representation**: The presentation of results and metrics could benefit from adjustments for physical interpretability.

    - I suggest adjusting the x-axis in Figures 2 and 3 to reflect physical simulation time instead of “time scope” or “time frequency,” as these concepts are not well-defined in computational chemistry.

    - Figure 2b shows only marginal improvement over EGNN. Could the authors elaborate on why this is the case? Were there examples where the proposed method underperformed compared to other methods?

    - Figure 2 reports the Mean Squared Error (MSE) of predicted positions, which is a common ML metric. However, it would be more informative to verify whether the method produces physically consistent MD trajectories. For example, the MD17 dataset was generated under NVT or NVE ensembles, which conserve certain physical observables over time. I suggest the authors compute the energies of their predicted atomic positions to assess if the method preserves energy conservation. If energy conservation is not maintained, how does this method compare with others in this respect? I believe this approach would provide a more physically sound measure of prediction accuracy.

---

### Official Review · Reviewer_Wcck · 2024-11-03

**Soundness:** 2
**Presentation:** 3
**Contribution:** 2
**Rating:** 3
**Confidence:** 5

**Summary:**

The paper introduces a neural operator model that leverages the graph fourier transformation (GFT) to predict the time evolution of atomic systems in molecular dynamics. The approach performs graph fourier decomposition of the invariant and equivariant features of a molecular graph, and then learns a neural ODE for each fourier component to predict the time evolution. Finally, it decodes the structure with inverse GFT and applies the method iteratively to propagate over time. The method is evaluated on MD17 dataset and benchmarked against standard graph neural ODE approaches and one-step equivariant prediction approach.

**Strengths:**

- The approach is inspired strongly from physics. The authors use the relationship between spatial and temporal evolution to argue the advantage of modeling time evolution of different spatial fourier components separately. The introduction is well written and the approach is well motivated.
- The paper performs ablation study and detailed analysis over the long-term evolution and correlation between different spatial features.

**Weaknesses:**

- The empirical results are too weak to demonstrate the usefulness of the approach. In Table 1, the performance of their approach is very close to EGNN, an equivariant network without explicitly modeling the time propagation. EGNN also doesn’t model the graph fourier components explicitly. The results indicate that both the neural operator and fourier features are not contributing significantly to the prediction of long-term evolution of atomic structure.
- The long-term prediction stability results in Figure 2 are not very meaningful. All predicted structures deviate from the original structure by > 0.5 A in terms of MSE. This is much larger than typical movement of atoms inside a molecule. It either indicates the model breaking apart or the drift of molecules from the origin.
- The method is only evaluated in MD17, the simplest dataset for molecular dynamics. To demonstrate the practical usage of the approach, the authors should consider more complex datasets [1] like Alanine dipeptide, water, etc.

[1]  Fu, Xiang, et al. "Forces are not enough: Benchmark and critical evaluation for machine learning force fields with molecular simulations." arXiv preprint arXiv:2210.07237 (2022).

**Questions:**

- What is the MSE of a random molecule in MD17 compared with the initial structure? This should provide an upper bound for the results in Table 1.

---

### Official Review · Reviewer_CcHG · 2024-11-03

**Soundness:** 3
**Presentation:** 1
**Contribution:** 3
**Rating:** 5
**Confidence:** 5

**Summary:**

The paper proposes an interesting architecture to model multi-scale spatial and temporal dynamics of molecules, combining equivariant GNN, graph Fourier transform and neural ODE. The idea is enticing. But I find the execution lacking rigor and appropriate analysis. The paper has a lot of potential, but the present form is not yet convincing. Note I gave a rating of 1 on presentation. I found the details confusing. Please improve!

**Strengths:**

The idea of multiscale modeling using GFT is physically motivated. The authors employed several cutting-edge techniques. The application area of MD is an very high-impact area.

**Weaknesses:**

Three big complaints:

* I have tons of questions about the technical details (see questions below).

* Lack of more meaningful metrics in the context of an MD surrogate model: physics metrics rather than point-wise MSE. Over long time horizons the atom wise MSE is irrelevant. Trajectories always diverge. What about bond length, bond angle, total energy... distributions? Those are far more important than MSE.

* Please state the intended application scope of this work (details below).

**Questions:**

* Please correct "ahallengeforementioned"

* Note some recent developments in ML-based surrogate models of molecular dynamics such as Two-for-One (Arts et al 2023), Score dynamics (Hsu et al 2024) and F3low (Li et al 2024)


* What about transferability? Can a model trained on one dataset be used in another? For this paper I am not demanding proof of transferability. But such discussions are necessary.

* The proposed method is deterministic. Can stochasticity be considered/treated?

* Closely related to the above: What's the nature of the MD reactions being modeled? It is downhill relaxations? Then a deterministic model might work. Or thermal fluctuations and barrier crossing? Then this work won't apply. The authors **should** clearly state the intended application area of their method.

* Are latent feature vectors h and \tilde{h} scalars (rotationally invariant), and x and \tilde{x} vectors? If so, consider stating this explicitly, as \tilde{h} and \tilde{x} are really just latent features of different irreps (0e and 1o) and it is more appropriate to call them so.

* I suggest distinguishing between input features h and latent features \tilde{h}, if that's the correct way to understand it. Please explain what's h.

* I find eq (6) confusing. Why the sum of real coordinates and a vectorial (1o) latent feature. It makes sense to me to do a Fourier transform on vectorial (and scalar) latent features, but including real coordinates x in the GFT looks odd to me, because then you'd have to worry about translational invariance. Please explain how the center of mass (k=0 mode of FT of coordinates) is treated in this work. Physically, because of translational invariance, the k=0 mode shouldn't matter. Otherwise, it makes more sense to me to have the eq(6) without adding real coordinates x.

* Eq (15): It is confusing. Eq (5, 6, 7) says the input features h are fed into the GNN, and \tilde{h} is the output rather than input.

* Does eq (15) suggest the dynamics modeled is time-dependent dynamics? Also in eq (10,11), they are time dependent ODE. Was the ODE explicitly time dependent?

* Eq (7, 16): the term "updated" is confusing. I'd reserve "update" to mean changes from one timestep to the next, i.e. temporal updates. I understand it's meant in a computer science context. But talking about dynamics, it's better to avoid confusion.

* eq (16, 17): are the LHS  h and x, rather than the tilde ones? From Eq (18) it appears the tilde symbols were meant to mean both latent features and model predictions. Please rephrase. Strictly speaking eq (16, 17) are just wrong: eq (16) looks like a fixed point evaluation problem but apparently that's not the intention. Eq. (17) similarly means literally the updates are zero on RHS.

* Are the learnable models such as phi and chi in the encoder and decoder the same? If not, please use different symbols.

* How is \Delta T (line 377) defined. Is the the uniform timestep between t1, t2, ...? What's the MD timestep? And please also express \Delta T in real units like ps.

* Table 1: how irregular are the timesteps? How is the related to \Delta T?

* Fig. 2: again please state clearly how much is a timestep. Consider adding a "real-time" x-axis with ps units on the plot.

* Fig. 2: My biggest complaint of this paper is the lack of more meanningful metrics as a MD surrogate model: physics metrics rather than point-wise MSE. Over long time horizons the point wise MSE is irrelevant. What about bond length, bond angle, total energy... distributions? Those are far more important than MSE.

---

### Public Comment · ~Fang_Sun3 · 2025-03-14
**Author information**

Fang Sun,
Address: 405 Hilgard Avenue, Engineering Building VI,
Email: fts@cs.ucla.edu.
Cellphone: 310-572-7232
Any questions regarding the paper should be directed to this contact.

---

### Meta-Review · Area_Chair_wz69 · 2024-12-19

**Metareview:**

The reviewers all agreed that the methodology presented in the paper is not fully developed and therefore they are leaning reject. The authors made a serious and appreciated effort to convince the reviewers. However, this did convince those reviewers who answered.

Rejection is therefore recommended with an encouragement to use the feedback from the reviewers into account when working on a revision.

**Additional Comments On Reviewer Discussion:**

None.

---

### Decision · Program_Chairs · 2025-01-22

Reject